# Factors Related to Cardiac Troponin T Increase after Participation in a 100 Km Ultra-Marathon

**DOI:** 10.3390/diagnostics10030167

**Published:** 2020-03-19

**Authors:** Łukasz A. Małek, Anna Czajkowska, Anna Mróz, Katarzyna Witek, Dariusz Nowicki, Marek Postuła

**Affiliations:** 1Department of Epidemiology, Cardiovascular Disease Prevention and Health Promotion, National Institute of Cardiology, 04-635 Warsaw, Poland; 2Faculty of Tourism and Recreation, University of Physical Education in Warsaw, 00-809 Warsaw, Poland; anna.czajkowska@awf.edu.pl (A.C.); dariusz.nowicki@awf.edu.pl (D.N.); 3Faculty of Physical Education, University of Physical Education in Warsaw, 00-809 Warsaw, Poland; anna.mroz@awf.edu.pl (A.M.); katarzyna.witek@awf.edu.pl (K.W.); 4Department of Experimental and Clinical Pharmacology, Medical University of Warsaw, 02-097 Warsaw Poland; mpostula@wum.edu.pl

**Keywords:** running, exercise, marathon, troponin, risk factor

## Abstract

Background: Intensive and prolonged exercise leads to a rise of troponin concentration in blood. The mechanism responsible for troponin release during exercise remains ill-defined. The study aim was to search for risk factors of troponin increase after a prolonged endurance competition. Methods: The study included a group of 18 amateurs, healthy volunteers (median age 41.5 years, interquartile range – IQR 36–53 years, 83% male) who participated in a 100 km running ultra-marathon. Information on demographic characteristics, pre- and post-race heart rate, blood pressure, body composition and glucose, lactate (L), troponin T (hs-TnT) and C reactive protein (hs-CRP) concentration were obtained. Additionally, data on L and glucose levels every 9.2 km and fluid/food intakes during the race were collected. Results: There was a significant hs-TnT increase after the race exceeding upper reference values in 66% of runners (from 5 IQR 3–7 ng/L to 14 IQR 12–26 ng/L, *p* < 0.0001). None of the baseline parameters predicted a post-race hs-TnT increase. The only factors, correlating with changes of hs-TnT were mean L concentration during the race (rho = 0.52, *p* = 0.03) and change of hs-CRP concentration (rho = 0.59, *p* = 0.01). Conclusions: Participation in a 100 km ultra-marathon leads to a modest, but significant hs-TnT increase in the majority of runners. Among analysed parameters only mean lactate concentration during the race and change in hs-CRP correlated with troponin change.

## 1. Introduction

Regular physical activity leads to many health benefits including cardio-protective activity [1,2]. Recent guidelines of the European Society of Cardiology have increased the recommended weekly volume of moderate to vigorous exercise from 150 min to 210–420 min as optimal cardiovascular prevention [3]. However, there is a group of athletes who far exceed these recommendations by engaging in long-lasting training and ultra-endurance competitions. The effects of these extreme forms of exercise on health are much less studied [2].

Bouts of intensive or long-lasting exercise (such as marathons or ultra-marathons) have been shown to increase blood levels of cardiac troponin, a known selective marker for myocardial injury and a major component of a current diagnosis of myocardial infarction [4,5,6,7]. Troponin rise after exercise is usually discrete with levels returning to reference values within hours [8] and may be accompanied by an increase of natriuretic peptide levels and transient decrease of left or right ventricular systolic function without long-term consequences [9,10,11,12]. In line with that, recent studies with new imaging techniques demonstrated that intensive endurance training does not seem to promote myocardial fibrosis [6,13,14,15].

It is believed that cardiac troponin released after endurance exercise comes from the cytosolic pool and does not signify injury to contractile parts of the cardiomyocytes [4,7]. However, the exact mechanism explaining troponin rise related to physical activity remains ill-defined [4,7]. It has been previously related to athletes age and experience, exercise duration and intensity potentially influencing dehydration, inflammation or pH imbalance during the exercise [4,7]. More detailed knowledge of the risk factors of this form of troponin increase could help in understanding the mechanism behind this phenomenon. They may also improve differential diagnosis in case of post-exercise suspicion of the coronary events in athletes [5,7].

Therefore, the study aimed to search for risk factors of troponin increase after participation in a 100 km ultra-marathon in middle-age, amateur healthy runners. 

## 2. Materials and Methods

### 2.1. Subjects and Design

The study was conducted during a 100 km running ultra-marathon on flat terrain (asphalt, bitumen track and short parts of cobblestone), which took place on 10th November 2018 at the University of Physical Education in Warsaw (www.supermaraton100lecia.pl). The race consisted of 65.10 laps of 1535.89 m distance and was accredited by the Polish Athletics Association as National Championships on 100 km. 

Out of 204 runners taking part in the race, we included 18 amateurs, healthy runners (3 females) who volunteered to participate in the study and to follow the whole protocol of the study. 

Each study participant underwent initial screening in the form of a medical questionnaire to exclude any known medical conditions. It was followed by the assessment of (1) body composition including body mass, total body water (TBW), body fat (FAT) and free fatty mass (FFM), (2) resting heart rate (HR) and blood pressure (BP) measurement and (3) blood draw from an antecubital vein for baseline analysis of high-sensitivity troponin T (hs-TnT) and high sensitivity C-reactive protein (hs-CRP) levels. At the same time analysis of baseline capillary lactate (L) and glucose (Glu) concentration was performed. 

During the race, we collected data on fluid and food intake. Additionally, every 6 laps (approximately every 9.2 km) runners had fingertip capillary L and Glu assessment. Immediately after the end of running each participant underwent a final assessment of capillary L and Glu concentration followed by venous blood draw for hs-TnT and hs-CRP assessment and BP, HR and body composition analysis. In collaboration with a certified company (datasport.pl) we have collected data on the time of the race, mean pace and total distance covered by each runner participating in the study. 

### 2.2. Methodology

Body composition was analysed with means of a bio-impedance device (Tanita BC 41 MA, Tanita Inc, Tokyo, Japan). To assess hs-TnT and hs-CRP levels an electrochemiluminescence immunoassay method (ECLIA) on Roche Cobas e411 analyser (Roche Diagnostics, Mannheim, Germany) was used. Reference values for this fourth-generation hs-TnT assay were < 14 ng/L and < 5 mg/dL for hs-CRP assay. Fingertip capillary L and Glu assessment were performed with Biosen C-Line Glucose and Lactate analyser (EKF Diagnostics, Cardiff, United Kingdom).

### 2.3. Study End-Point

The study end-point was a change of hs-TnT concentration, assessed immediately after participation in a running ultra-marathon in comparison to baseline hs-TnT value. 

### 2.4. Statistical Analyses

All results for categorical variables were presented as number and percentage. Continuous variables were expressed as median and interquartile range (IQR). Wilcoxon test for paired samples or Mann-Whitney test for unpaired samples were applied to compare continuous data. To assess the correlation between continuous variables, the Spearman test was applied. All tests were two-sided with the significance level of *p* < 0.05. Statistical analyses were performed with MedCalc statistical software 10.0.2.0 (MedCalc, Mariakerke, Belgium). 

### 2.5. Ethical Considerations

The study had an approval of the Ethics Committee of the Regional Medical Chamber in Warsaw (no 52/17), with written informed consent obtained from all participants.

## 3. Results

### 3.1. Baseline Characteristics

Baseline and running characteristics of the runners participating in the study are presented in Table 1. Nine runners (50%) have not participated in any ultra-marathon before. Of those 6 have completed at least one marathon and remaining 3 have run only distances up to 30 km. Eight runners (44%) completed the full distance of 100 km within the time limit of 12 h. Other 10 participants run between 52 and 91 km (median 74 km, IQR 71–89). The median time of running in the studied group was 10.3 h (IQR 8.3–11.5) and the median pace was 8.7 min/km (IQR 8.0–9.4).

### 3.2. Pre- Vs. Post-Race Values

At baseline, the troponin level was below the upper reference limit (99th percentile) in all runners. In all participants the troponin concentration increased after the race (from 5 IQR 3–7 ng/L ng/L to 14 IQR 12–26 ng/L, *p* < 0.0001). It exceeded upper reference values in 66% of runners. Maximal post-race hs-TnT concentration was 38 ng/L (Figure 1).

Table 2 presents the comparison of pre- and post-race values of other parameters. All participants had higher systolic and diastolic BP and lower HR before the race in comparison to post-race values. Running led to a decrease in body mass caused by TBW and FFM loss without significant FAT decrease. During the race, participants drank 1950 mL (IQR 1200–2600 mL) of fluids and ingested 1500 kcal (IQR 919–2340 kcal). There was an increase of CRP after the race, with normal values in all runners before the race and values exceeding the reference values in 6 runners (33%) after the race. 

### 3.3. Factors Correlating with Troponin Increase

Subsequently, we have analysed, which of the parameters correlated with the change of hs-TnT concentration (Table 3). Interestingly, this was not the case for any of the analysed baseline demographic, clinical and biochemical parameters. We also did not find any correlation between the delta of hs-TnT and the time of running, running pace, fluid and food intake during the race, changes in body composition or pre-post race changes in L and Glu concentration. The only factor correlating with the change of hs-TnT was the change of hs-CRP. 

### 3.4. Periodically Assessed Parameters and Troponin Increase 

There was no significant change between L values before in comparison to after the race. However, L concentration fluctuated during the race in all runners, with an example demonstrated in Figure 2 affecting mean L concentration during the race rather than post-race L values. 

Runners with significant hs-TnT increase after the race had more L peaks > 2 mmol/L during the race (median 7 IQR 4–8 vs. 3.5 IQR 2–5, *p* = 0.04). In consequence, those with mean L concentration during the race above the median of 2.1 mmol/L had a significantly higher increase of hs-TnT (15 IQR 11–19 vs. 8 IQR 7–12 vs. ng/L, *p* = 0.02) as demonstrated on Figure 3. Mean L concentration during the race correlated with change in hs-TnT (rho = 0.52, *p* = 0.03).

At the same time, this was not the case for another periodically assessed parameter - glucose. Glucose fluctuations during the race and mean Glu concentration during the race were unrelated to troponin changes.

## 4. Discussion

We have demonstrated that participation in a 100 km marathon leads to a significant increase of hs-TnT beyond upper reference limits in 66% of runners. High-sensitivity troponin T measured with a fourth-generation assay, as in our study, is a cardio-selective marker and its increase is believed to have purely cardiac and not skeletal muscle origin [4]. However, it should be noted, that maximal values of post-race hs-TnT in studied runners remained within low probability range of acute coronary syndrome suspicion [5].

Analysis of risk factors of cardiac troponin T increase showed that mean L concentration during the ultra-marathon and hs-CRP increase were the only parameters correlating with troponin T rise after the race. Previous studies have also demonstrated increased hs-CRP levels after prolonged exercise, while L concentrations post-exercise remained unchanged or rose, depending on the physical activity intensity [16,17,18]. However, lack of pre- and post-exercise difference in L concentration does not exclude fluctuations of L levels during the competition, as in our study, which may affect mean L concentration. Periodic L sampling during long-lasting endurance competitions is rarely performed in clinical studies due to logistic challenges. Therefore our findings regarding fluctuations in lactate concentration and their relation to increased troponin concentration should be considered as a novel finding, not previously reported in the literature. 

The exact mechanism connecting lactate fluctuations and hs-CRP to hs-TnT increase is not known. Long-lasting endurance event is usually performed around (or even below) first ventilator threshold. However, periods of anaerobic metabolism during the race may lead to an increase in lactate concentration [18,19]. We postulate that periodic changes in lactate concentration may reflect systemic changes in acid-base and electrolytic balance as described in other studies [18,19,20,21]. As shown by our and other research studies, ultra-endurance exercise promotes inflammatory reaction measured in our study by the change of hs-CRP concentration [4,7]. Rise of hs-CRP may be a part of an acute-phase reaction to muscle damage mediated by cytokine system, mainly Il-6, or by lactate increase [16,17,18]. Acid-base or electrolyte alterations and inflammatory reaction may affect cardiomyocyte cell membrane by increasing its permeability [4,7]. Finally, increased permeability of the cardiomyocyte cell membrane may be responsible for the leakage of hs-TnT from the cytosol to the blood and mild hs-TnT increase observed immediately after the race [4,7]. In line with that, previous research showed a relation between inflammatory markers (including hs-CRP) and troponin increase post-exercise [22,23]. We are aware of the fact that this line of reasoning should be considered as a proposal for future more detailed research, as we were unable to confirm all steps of this hypothesis. In particular, increased cardiomyocyte cell membrane permeability may be also caused by mechanical stress or by oxygen radicals [4,7].

We did not find any correlation between troponin increase and some other previously described risk factors such as younger age, running experience (assessed by weekly covered distance in our study), running intensity (measured with pace and running time) or dehydration [4,7]. It could be caused by the homogeneity of the studied group, which included mainly mid-age, amateur runners with similar experience and comparable race characteristics. Also, unlike in some other studies, none of the analysed baseline biochemical parameters predicted the outcome [24].

Our study has some limitations. First of all, it was conducted on a relatively small number of runners, which precluded any multivariable analysis. However, specifics of the study protocol with a collection of data on fluid and food intake and capillary blood sampling for lactic acid and glucose every 9.2 km with all its logistic challenges made the inclusion of a larger group of participants impossible. We did not want to significantly influence the time of running of the participants, by not causing longer than necessary delays for data collection and blood sampling. For the same reason, we could not include semi-professional or professional runners participating in National Championships of 100 km. Therefore, we focused on amateur athletes only, which could have potentially influence the results. Furthermore, we were unable, as in other similar studies, to correct the post-race biochemical results for the potential changes in plasma volume after long-distance running, which could have affected the results. Finally, we did not analyse the role of potentially interesting cardio-vascular biomarkers such as microRNAs, ST2 protein and others, which could have shed new light on the increase of troponin concentration post-exercise [25]. Nevertheless, we believe that this does not diminish the value of the main findings.

Our results also show that the degree of cardiac troponin T concentration increase after an extreme ultra-marathon remains in the range typical for low cardiovascular risk of an acute event. This can help in the differential diagnosis in this group of patients if deemed necessary [5,7]. Finally, presented observations may help to explain the paradox between cardiac troponin increase after the endurance physical activity and lack of permanent myocardial injury in life-long elite runners observed in other studies [4,6,7,13,14,15]. As the increase in troponin concentration even after an extremely long race is very mild is it unlikely that it can lead to significant myocardial fibrosis. 

## 5. Conclusions

Participation in a 100 km ultra-marathon leads to a modest, but significant high-sensitivity troponin T increase in the majority of runners. Among analysed parameters only mean lactic acid concentration and change of high-sensitivity C reactive protein correlated with the change of troponin level.

## Figures and Tables

**Figure 1 diagnostics-10-00167-f001:**
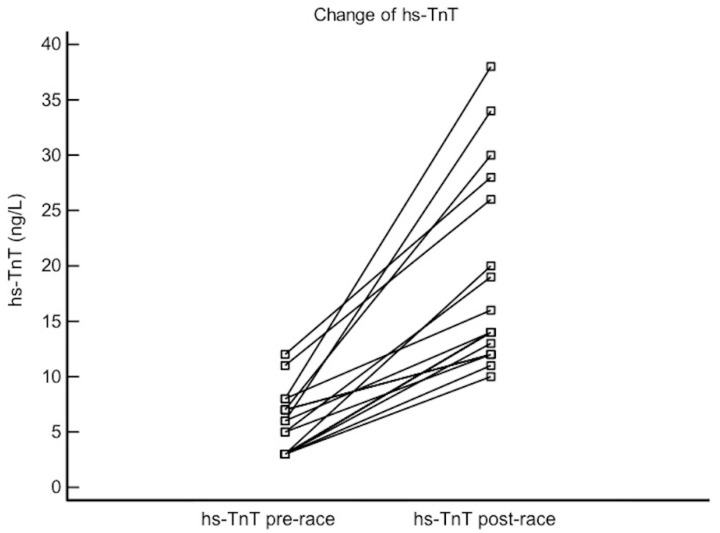
High-sensitivity troponin T (hs-TnT) concentration before and immediately after the end of running.

**Figure 2 diagnostics-10-00167-f002:**
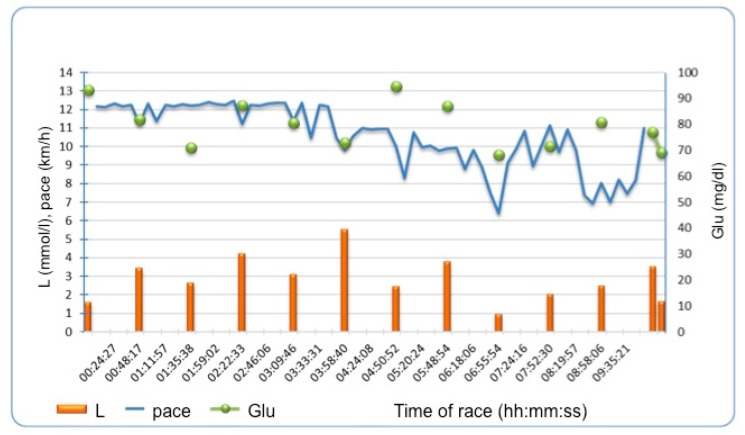
Example of lactate and glucose fluctuations during the race and running pace in one of the runners. L—lactate, Glu- glucose.

**Figure 3 diagnostics-10-00167-f003:**
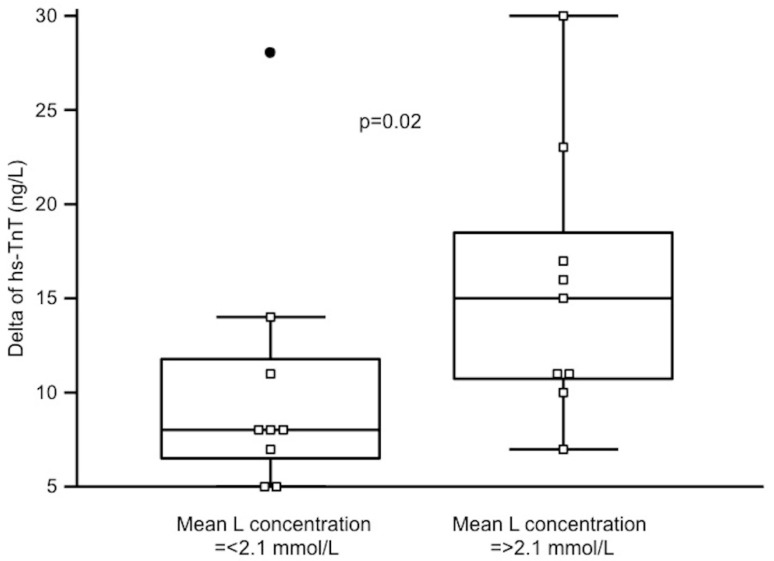
Changes of high-sensitivity TnT (hs-TnT) in runners with mean lactate (L) concentration during the race below and above the median. A black dot on the left is an outstanding result in one of the runners.

**Table 1 diagnostics-10-00167-t001:** Baseline and running characteristics of the studied group.

Parameter	Ultra-Marathon Runners *n* = 18
Male sex (%)	15 (83)
Age, yrs (IQR)	41.5 (36–53)
BMI, kg/m2 (IQR)	24.6 (22.7–25.7)
Years of running (IQR)	4.3 (3.5–6.0)
Years of ultra running (IQR)	2 (0–3)
Weekly running distance, km (IQR)	67.5 (40–85)
Number of ultra races completed (IQR)	2 (0–10)
Longest completed race, km (IQR)	58 (42–80)

Data are presented as number and percentage or median and interquartile range (IQR).

**Table 2 diagnostics-10-00167-t002:** Pre- and post-race values of the analysed parameters.

	Pre-Race *n* = 18	Post-Race *n* = 18	*p*
Body mass, kg	75.0 (69.6–83.3)	74.2 (69.7–82.7)	<0.0001
TBW, kg	44.2 (39.8–49.9)	41.6 (39.2–46.7)	0.008
TBW, %	58.1 (54.4–60.1)	56.6 (54.2–60.1)	0.32
FAT, kg	10.9 (7.9–12.5)	10.5 (7.6–12.2)	0.46
FAT, %	13.7 (11.4–17.5)	14.2 (10.7–18.6)	0.73
FFM, kg	65.7 (61.2–72.0)	64.1 (60.6–70.7)	0.0001
FFM, %	86.3 (82.5–88.7)	85.8 (81.4–89.3)	0.77
HR, bpm	54.5 (50–60)	81.5 (76–93)	<0.0001
SBP, mmHg	137 (130–146)	123 (109–133)	0.0004
DBP, mmHg	84 (92–91)	73 (70–78)	<0.0001
CRP, mg/dL	0.7 (0.43–1.1)	3.2 (1.9–8.1)	<0.0001
hs-TnT, ng/L	5 (3–7)	14 (12–26)	<0.0001
L, mmol/L	2.0 (1.7–2.4)	2.2 (1.4–3.5)	0.22
Glu, mg/dL	89 (86–95)	93 (80–100)	0.83

DBP—diastolic blood pressure, FAT—body fat, FFM—free fatty mass, Glu—glucose, HR—heart rate, hs-CRP—high-sensitivity C reactive protein, L—lactate, SBP—systolic blood pressure, TBW—total body water, hs-TnT—high-sensitivity troponin T.

**Table 3 diagnostics-10-00167-t003:** Correlation between change in hs-TnT and analysed parameters.

	*hs-TnT Change*	
	rho	*p*
**Baseline Parameters**		
Age, yrs	0.04	0.86
Weekly distance covered, km	0.23	0.34
Body mass pre, kg	0.34	0.16
TBW pre, kg	0.17	0.49
FFM pre, kg	0.31	0.20
HR pre, bpm	0.01	0.96
SBP pre, mmHg	0.04	0.86
DBP pre, mmHg	0.01	0.99
L, mmol/L	0.11	0.66
Glu, mg/dL	−0.26	0.28
Hs-CRP, mg/dL	0.17	0.49
Hs-TnT, mmol/L	0.23	0.32
**Race Parameters and Post-Race Changes**		
Race time, hours	0.45	0.18
Pace, min/km	0.12	0.62
Water intake during the race, mL	−0.04	0.87
Food intake during the race, kcal	0.07	0.76
Delta body mass, kg	0.08	0.74
Delta TBW, kg	0.31	0.20
Delta FFM, kg	0.29	0.23
Delta L, mmol/L	0.30	0.21
Delta Glu, mg/dL	0.04	0.85
Delta hs-CRP, mg/dL	0.59	0.01

DBP—diastolic blood pressure, FFM—ree fatty mass, Glu—glucose, HR- heart rate, L—lactate concentration, SBP—systolic blood pressure, TBW—total body water, hs-TnT—high-sensitivity troponin T.

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
