# Peer review of "Factors Related to Cardiac Troponin T Increase after Participation in a 100 Km Ultra-Marathon"

_diagnostics, 2020, doi:10.3390/diagnostics10030167_

Round 1
Reviewer 1 Report
General comments:
This is an interesting study of an experienced research group in the field, which confirms previously published results. The novel and difficult to perform design is lactate measurement during running. However, the main problem with this and several other similar studies is that troponin concentrations are not corrected for the potential changes in plasma volume after long-distance running, e.g. cardiac Troponin T should be given /g of total protein or using hemoglobin or hematokrit values for correction. The correlation of changes in the 2 tested plasma proteins troponin and CRP supports this need.
The differences of troponin post-race values stratified by the median of the average lactate capillary blood concentrations during race should be given as numbers in the text, or as a table or figure to allow to assess the absolute differences in concentrations, i.e. whether this difference - although significant - may be within the analytical variation of the assay.
Reviewer 2 Report
Dear editor,
About the article entitled “Factors related to cardiac troponin T increase after participation in a 100 km ultra-marathon”, enclosed my suggestions:
First, I would like to remember the importance of the assay of biomarkers at baseline (and before an intervention) to predict the worse prognosis also in patients without cardiovascular disease (Orthopedics. 2017 May 1;40(3):e417-e424. doi: 10.3928/01477447-20170109-02). Please discuss this point, the reference could help you.
Secondly, parallely to cardiac markers of damage as troponin, other markers of inflammation (Curr Pharm Des. 2020 Feb 13. doi: 10.2174/1381612826666200213123029) could be indicated to evaluate the cardiac performance and the prognosis of studied patients. Do you have these data in your study? Could give me this feedback? Please response to this point.
Again, I would like to know data about cardiac chambers volumetry, systolic and diastolic volumetry and ejection fraction. Do you have these data? Do they correlate with troponin?
What was the arrhythmic burden of the study population?
Please include a clear description about Study endpoints as a paragraph.
Please include a description about Study Limitations. It is important.
I see low quality of figures and tables.
Improve the number and novelty of references.
Author Response
Dear editor,
About the article entitled “Factors related to cardiac troponin T increase after participation in a 100 km ultra-marathon”, enclosed my suggestions:
First, I would like to remember the importance of the assay of biomarkers at baseline (and before an intervention) to predict the worse prognosis also in patients without cardiovascular disease (Orthopedics. 2017 May 1;40(3):e417-e424. doi: 10.3928/01477447-20170109-02). Please discuss this point, the reference could help you.
Thank you for this comment. We did not find any correlation between baseline values of biomarkers (L, glucose, hs-CRP or TnT) and change of troponin T after the race. We have added this information to the text and also to table 3: “Interestingly, this was not the case for any of the analyzed baseline demographic, clinical and biochemical parameters.”We have also included a new citation of the proposed manuscript in the discussion section: “Also, unlike in some other studies, none of the analyzed baseline biochemical parameters predicted the outcome24.”
New rows in Table 3.
|
|
hs-TnT change |
|
|
|
rho |
p |
|
Baseline parameters |
||
|
L, mmol/L |
0.11 |
0.66 |
|
Glu, mg/dL |
-0.26 |
0.28 |
|
Hs-CRP, mg/dL |
0.17 |
0.49 |
|
Hs-TnT, mmol/L |
0.23 |
0.32 |
Secondly, parallely to cardiac markers of damage as troponin, other markers of inflammation (Curr Pharm Des. 2020 Feb 13. doi: 10.2174/1381612826666200213123029) could be indicated to evaluate the cardiac performance and the prognosis of studied patients. Do you have these data in your study? Could give me this feedback? Please response to this point.
We have only analyzed one inflammatory marker – hs-CRP, which is included in the manuscript. Other inflammatory markers could have parallel the changes in hs-CRP, but not necessarily, and therefore we have added this point to the limitations section along with the proposed citation: “Finally, we did not analyze the role of potentially interesting cardio-vascular biomarkers such as microRNAs, ST2 protein and others, which could have shed new light on increase of troponin concentration post-exercise25.”
Again, I would like to know data about cardiac chambers volumetry, systolic and diastolic volumetry and ejection fraction. Do you have these data? Do they correlate with troponin? What was the arrhythmic burden of the study population?
All of the participants declared themselves as healthy without any significant medical history. Physical examination did not reveal any abnormalities. All of the participants were active runners, without any declared symptoms during or off exercise. However, we did not perform echocardiography or any other imaging study to analyze cardiac chamber size or ventricular function. We also did not perform Holter ECG to analyze the arrhythmic burden of the studied group.
Please include a clear description about Study endpoints as a paragraph.
We have included the following information as a separate paragraph entitled Study endpoint: “The study end-point was a change of hs-TnT concentration, assessed immediately after participation in a running ultra-marathon in relation to baseline hs-TnT value..“
Please include a description about study limitations. It is important.
Description of study limitations has been present at the end of the discussion section. We have extended in further to sound: “Our study has some limitations. First of all, it was conducted on a relatively small number of runners, which precluded any multivariable analysis. However, specifics of the study protocol with collection of data on fluid and food intake and capillary blood sampling for lactic acid and glucose levels every 9.2 km with all its logistic challenges made inclusion of larger group of participants impossible. We did not want to significantly influence the time of running of the participants, by not causing longer than necessary delays for data collection and blood sampling. For the same reason we could not include semi-professional or professional runners participating in National Championships of 100 km. Therefore, we focused on amateur athletes only, which could have potentially influence the results. Furthermore, we were unable, as in other similar studies, to correct the post-race biochemical results for the potential changes in plasma volume after long-distance running, which could have affect the results. Finally, we did not analyze the role of potentially interesting cardio-vascular biomarkers such as microRNAs, ST2 protein and others, which could have shed new light on increase of troponin concentration post-exercise25. Nevertheless, we believe that this does not diminish the value of the main findings.”
I see low quality of figures and tables.
The quality of the figures might have been low due to their embedding within the text in the Words document. They have much better resolution as separate files, which we have attached.
Improve the number and novelty of references.
Most of the references are from last 10 years except some pivotal/milestone studies in the area from earlier years. We have improved the number and novelty of the references wherever it was possible, also by adding the 2 references suggested by the reviewer from 2017 and 2020.
Round 2
Reviewer 1 Report
Unfortunately the main limitation, i.e. no correction of laboratory test results post-race for plasma volume changes, could not be corrected retrospectively. This has been included as a study Limitation.
Author Response
Thank you for this comment. We are aware of this limitation and we have discussed it in the limitation section.
Reviewer 2 Report
The authors improved the quality of the article by a point by point revision of the manuscript.
Author Response
We have performed an English spell and language check - highlighted in red in the text. We have found one newer publication to substitute ref. 7.